# LRPO: Enhancing Blind Face Restoration through Online Reinforcement Learning

## Abstract

Blind Face Restoration (BFR) encounters inherent challenges in exploring its large solution space, leading to common artifacts like missing details and identity ambiguity in the restored images. To tackle these challenges, we propose a **L**ikelihood-**R**egularized **P**olicy **O**ptimization (LRPO) framework, the first to apply online reinforcement learning (RL) to the BFR task. LRPO leverages rewards from sampled candidates to refine the policy network, increasing the likelihood of high-quality outputs while improving restoration performance on low-quality inputs. However, directly applying RL to BFR creates incompatibility issues, producing restoration results that deviate significantly from the ground truth. To balance perceptual quality and fidelity, we propose three key strategies: 1) a composite reward function tailored for face restoration assessment, 2) ground-truth guided likelihood regularization, and 3) noise-level advantage assignment. Extensive experiments demonstrate that our proposed LRPO significantly improves the face restoration quality over baseline methods and achieves *state-of-the-art* performance. The source codes and models are available at: https://anonymous.4open.science/r/LRPO-5874.

## 1 Introduction

Blind Face Restoration (BFR), which aims to reconstruct high-quality (HQ) faces from low-quality (LQ) inputs with unknown degradations, has made rapid progress in recent years. Modern BFR methods typically exploit various types of priors to establish direct mappings from LQ to HQ. These priors can be categorized into several types: (1) *geometric priors* (*e.g.*, facial landmarks (Chen et al., 2018; Kim et al., 2019), parsing maps (Chen et al., 2021), and component heatmaps (Yu et al., 2018)) that provide structural guidance; (2) *generative priors* (Wang et al., 2021; Chan et al., 2021; Yang et al., 2021) from pre-trained models like StyleGAN (Karras et al., 2019; 2020) that enable realistic detail reconstruction; (3) *discrete codebook priors* (Gu et al., 2022; Zhou et al., 2022) improve restoration fidelity; and (4) *diffusion priors* (Wu et al., 2024; Lin et al., 2024; Chen et al., 2024; Yue & Loy, 2024; Wang et al., 2023b) that have recently attracted significant attention. Diffusion models offer distinct advantages including robust generative capability, stable optimization, and superior control over output diversity, making them particularly effective for producing high-quality, visually pleasing face restorations.

However, despite the advantages of diffusion priors, BFR remains fundamentally challenging. The task is inherently an ill-posed inverse problem where a single LQ input can correspond to multiple plausible HQ solutions, making it difficult to determine the optimal restoration. Current methods are constrained by their deterministic nature–they learn a fixed one-to-one mapping that produces a single output without considering alternative solutions. This lack of exploration within the vast solution space prevents these methods from discovering potentially superior restorations, leading to suboptimal results (Zhou et al., 2021).

To address these exploration limitations, we propose incorporating reinforcement learning (RL) into BFR. RL has demonstrated remarkable success in expanding performance boundaries across various domains, particularly in language models (Shao et al., 2024; Yu et al., 2025) and vision models (Fan et al., 2023; Liu et al., 2025; Wang et al., 2025; Yuan et al., 2025), by enabling diverse exploration strategies rather than deterministic outputs. Building on recent advances that successfully integrate RL with diffusion models (Liu et al., 2025; Xue et al., 2025)–where the denoising process

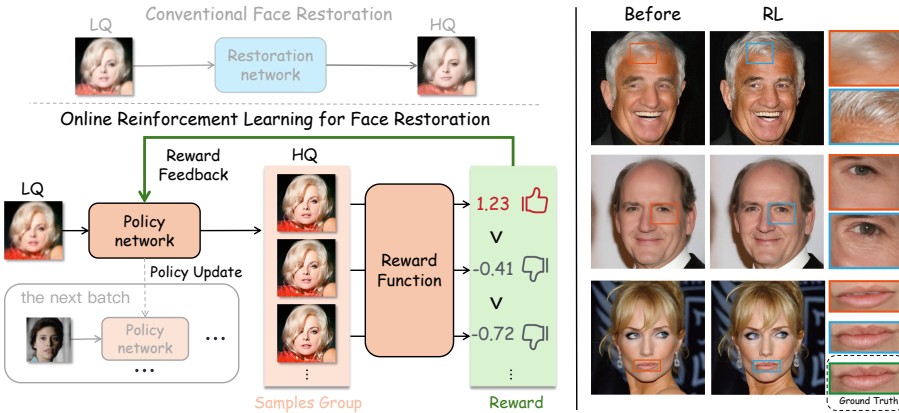

Figure 1: (Left) Our proposed online RL-based face restoration framework: an LQ face is input to the policy network $\pi_\theta$, which generates a group of HQ face candidates. The reward function evaluates each candidate and converts the scores into within-group relative advantages that guide policy optimization for the next iteration. The comparisons (Right) demonstrate the quality improvement achieved through RL optimization over the base model.

is formulated as a Markov decision process–we leverage RL's exploration capabilities alongside diffusion models' inherent randomness to systematically search BFR's solution space for optimal restorations that enhance both fidelity and perceptual quality.

To this end, we introduce the first online RL framework to BFR: Likelihood-Regularized Policy Optimization (LRPO). Our framework utilizes a policy network to generate multiple diverse HQ candidate faces from each LQ input, effectively exploring the solution space rather than following a single deterministic trajectory. As shown in Figure 1 (left), this multi-candidate sampling strategy allows systematic exploration of potential solutions. A reward function evaluates each candidate, and relative advantages computed among candidates from the same input guide the policy network optimization. Moreover, we introduce three key innovations to enable effective RL optimization: (1) We design a composite reward function that evaluates restoration quality by incorporating human preferences, perceptual quality, and fidelity metrics. (2) To prevent reward hacking (Skalse et al., 2022; Amodei et al., 2016)–where the policy exploits reward signals while deviating from authentic facial distributions–we implement ground-truth (GT) guided likelihood regularization that anchors the policy to the true data manifold. (3) We propose a noise-level advantage assignment mechanism that weights the advantages according to the importance of each denoising step, ensuring more effective policy updates.

In summary, our main contributions are as follows:

- We introduce online RL to BFR for the first time, modeling the learning process as exploration for superior restoration solutions. Specifically, we propose an LRPO framework that overcomes the limitations of single deterministic trajectory generation by exploring multiple restoration candidates through the RL training.
- We introduce three critical components for our proposed LRPO framework: a multi-faceted reward function that captures diverse restoration quality aspects, GT guided likelihood regularization that maintains authentic facial distributions while preventing reward exploitation, and adaptive advantage weighting that optimizes learning across different denoising stages.
- Our LRPO framework delivers substantial improvements in face restoration quality and establishes new *state-of-the-art* performance on standard evaluation metrics.

## 2 RELATED WORK

**Diffusion-based Blind Face Restoration.** Blind Face Restoration (BFR) aims to recover high-quality face images from these degraded inputs while preserving identity consistency and perceptual quality. Recently, diffusion models (Sohl-Dickstein et al., 2015; Ho et al., 2020) have gained popularity due

to their generative diversity and training stability. DR2 (Wang et al., 2023b) uses a diffusion model for degradation removal, followed by refinement through an enhancement module. DifFace (Yue & Loy, 2024) constructs a posterior distribution to map LQ images to HQ counterparts, utilizing the error-shrinkage property of pre-trained diffusion models for robust restoration. To accelerate training, LDM (Rombach et al., 2022) recommends training diffusion in the latent space. DiffBIR (Lin et al., 2024), built on LDM, employs ControlNet to guide restoration using low-quality faces as control signals. Some variations of diffusion models have been used for face restoration. InterLCM (Li et al., 2025) leverages the latent consistency model (LCM) to improve semantic consistency, restore images efficiently. FlowIE (Zhu et al., 2024) uses conditional rectified flow for faster inference with comparable restoration quality. However, diffusion models often suffer from poor identity consistency and loss of facial details in restored images. We find that intrinsic randomness of diffusion models can be leveraged through reinforcement learning with composite reward mechanisms to generate higher-quality, more detailed facial restorations.

**RL in Vision Generation.** Reinforcement learning has recently achieved remarkable success in improving large language model reasoning, particularly through policy gradient approaches such as PPO (Schulman et al., 2017) and GRPO (Shao et al., 2024). In the text-to-image (T2I) generation field, many methods have explored incorporating policy gradient approaches (*e.g.*, PPO) to align with human preferences. These methods explicitly cast diffusion denoising as a multi-step decision process and update the policy accordingly. DDPO (Black et al., 2023) improves alignment and aesthetics by optimizing rewards tied to human feedback. DPOK (Fan et al., 2023) studies online RL with Kullback–Leibler (KL) regularization on SD (Rombach et al., 2022). As a complementary approach to RL, Diffusion-DPO (Wallace et al., 2024) successfully adapts direct preference optimization to diffusion likelihoods, achieving significant improvements in human-preference alignment on SDXL (Podell et al., 2023). Beyond standard diffusion models, Flow-GRPO (Liu et al., 2025) is the first to bring GRPO into flow-matching model. Building on this, TempFlow-GRPO (He et al., 2025) further improves efficiency and stability by introducing trajectory branching and enabling process-level rewards without an intermediate reward model. Motivated by these successes, we present the first integration of policy gradient-based online RL into the BFR domain.

## 3 PRELIMINARY

**BFR Problem Modeling.** BFR is an ill-posed inverse problem. From a mathematical perspective, given a low-quality observation $c_{\text{LQ}}$, the posterior distribution of its corresponding high-quality $x_0$, denoted as $p(x_0|c_{\text{LQ}})$, has multiple feasible solutions (Menon et al., 2020). This posterior can be modeled as a mixture distribution:

$$p(x_0|c_{\text{LQ}}) = \sum_{k=1}^{K} w_k \cdot p_k(x_0|c_{\text{LQ}}) \tag{1}$$

Here, $K$ represents the number of possible high-quality solutions. Each distribution $p_k(x_0|c_{\text{LQ}})$ represents a real face distribution compatible with $c_{LQ}$, having a specific identity, expression, or detail, with a peak at $\mu_k$. The term $w_k$ denotes the probability of the $k$-th solution, where $\sum w_k = 1$. This multi-solution nature poses a challenge for existing methods. Without dedicated exploration mechanisms, restoration models tend to converge toward average solutions, resulting in characteristically blurry faces that lack rich textural details (Lugmayr et al., 2020).

**DDIM.** Diffusion models generate data through forward noise addition and reverse denoising processes. In the forward diffusion process, the data is progressively perturbed by Gaussian noise, defined as $q(x_t|x_{t-1}) = \mathcal{N}(\sqrt{\alpha_t}x_{t-1}, (1 - \alpha_t)\mathbf{I})$, where $\alpha_t \in (0, 1]$ controls the noise intensity, and $x_t$ represents the noisy data at time step $t$.

The denoising process recovers data via the conditional distribution $p_\theta(x_{t-1}|x_t, c)$, where $c$ is a condition. In the DDIM (Song et al., 2020) framework, the network predicts the noise $\epsilon_\theta(x_t, t, c)$. The one-step denoising formula (from $t$ to $t-1$) is given by:

$$\mu_\theta(x_t, t, c) = \sqrt{\alpha_{t-1}} \cdot \frac{x_t - \sqrt{1 - \alpha_t} \cdot \epsilon_\theta(x_t, t, c)}{\sqrt{\alpha_t}} + \sqrt{1 - \alpha_{t-1} - \sigma_t^2} \cdot \epsilon_\theta(x_t, t, c), \tag{2}$$

$$x_{t-1} = \mu_\theta(x_t, t, c) + \sigma_t \cdot \epsilon_t, \quad \epsilon_t \sim \mathcal{N}(\mathbf{0}, \mathbf{I}), \tag{3}$$

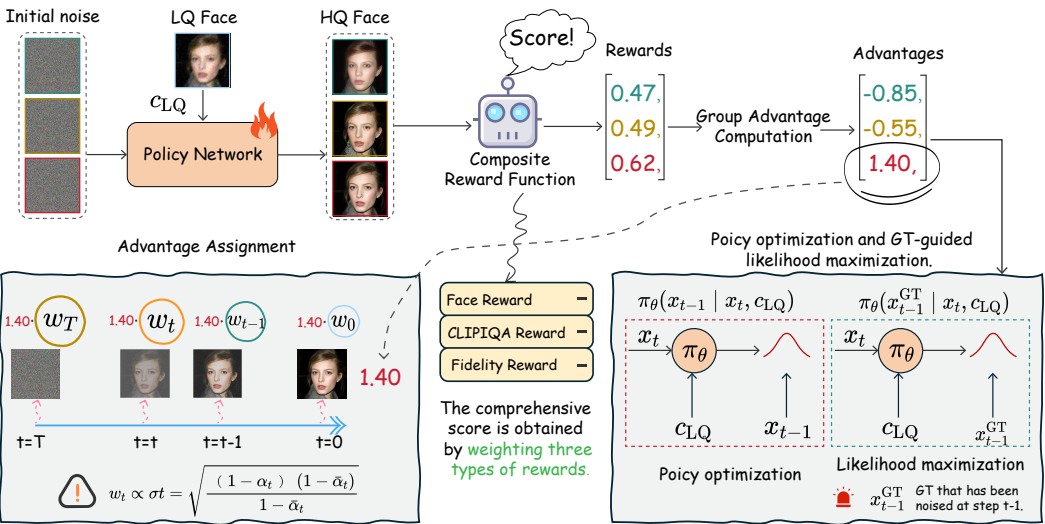

Figure 2: The overview of our proposed LRPO framework. The policy network produces multiple HQ restoration candidates from a single LQ input, which are then assessed by the reward function and transformed into advantage scores. The framework assigns weighted advantage scores to individual denoising steps according to their contribution to restoration quality, and integrates ground-truth guided likelihood regularization into the RL optimization objective to maintain fidelity.

where $\sigma_t = \eta \sqrt{\frac{1-\alpha_{t-1}}{1-\alpha_t} \cdot \left(1 - \frac{\alpha_t}{\alpha_{t-1}}\right)}$, and $\eta \in [0, 1]$ controls the randomness. When $\eta = 0$, the sampling is deterministic, producing a fixed generation path. When $\eta > 0$, random noise $\epsilon$ is introduced, bringing randomness.

Fan et al. (2023) proposes that DDIM can be formulated as a Markov Decision Process (MDP) defined by the tuple $(S, A, \rho_0, \pi, P, R)$. At time step $t$, the state is $\boldsymbol{s}_t \triangleq (\boldsymbol{c}, t, \boldsymbol{x}_t)$. The action is the denoised sample predicted by the model, $\boldsymbol{a}_t \triangleq \boldsymbol{x}_{t-1}$, with the policy defined as $\pi(\boldsymbol{a}_t|\boldsymbol{s}_t) \triangleq \pi_\theta(\boldsymbol{x}_{t-1}|\boldsymbol{x}_t, \boldsymbol{c})$. The state transition is deterministic, given by $P(\boldsymbol{s}_{t+1}|\boldsymbol{s}_t, \boldsymbol{a}_t) \triangleq (\delta_{\boldsymbol{c}}, \delta_{\boldsymbol{a}_t})$, where $\delta_{\boldsymbol{c}}$ denotes the Dirac distribution at $\boldsymbol{c}$. The initial state distribution is $P_0(\boldsymbol{s}_0) \triangleq (p(\boldsymbol{c}), \mathcal{N}(0, \mathbf{I}))$. The reward is provided only at the final step: $R(\boldsymbol{s}_t, \boldsymbol{a}_t) \triangleq r(\boldsymbol{x}_0, \boldsymbol{c})$. When $\eta > 0$, the DDIM supporting MDP can achieve reinforcement learning training.

## 4 METHODOLOGY

LRPO is an approach that enhances the BFR task using online RL, as shown in Figure 2. First, we initialize its policy network with an *off-the-shelf* diffusion-based face restoration model. Specifically, we employ DiffBIR (Lin et al., 2024) (our base model), a ControlNet-based (Zhang et al., 2023) approach that uses the LQ input as a control signal to guide restoration. Based on the GRPO (Shao et al., 2024) algorithm, we propose three core innovations for the BFR task: (1) We design a composite reward function that provides rewards for the diffusion denoising process. (2) We propose a ground-truth guided likelihood regularization term to penalize policy updates that deviate from real face data. (3) We develop a noise-aware advantage assignment mechanism to appropriately weight advantages based on denoising step significance.

### 4.1 GROUP RELATIVE POLICY OPTIMIZATION

The optimization goal of reinforcement learning is to maximize the expected cumulative reward. Unlike PPO (Schulman et al., 2017), GRPO (Shao et al., 2024) samples a group of $G$ trajectories $\{\{\boldsymbol{x}_T^{(i)}, \boldsymbol{x}_{T-1}^{(i)}, \dots, \boldsymbol{x}_0^{(i)}\}\}_{i=1}^G$, from the policy $\pi_{\theta_{\text{old}}}$, obtaining $G$ candidate reconstructions $\hat{\boldsymbol{x}}_0^{(i)}$,

which are decoded from $\boldsymbol{x}_0^{(i)}$ in latent diffusion, and their rewards $r^{(i)} = r(\hat{\boldsymbol{x}}_0^{(i)}, \boldsymbol{x}_{\text{GT}})$, where $\boldsymbol{x}_{\text{GT}}$ denotes the ground truth corresponding to $\hat{\boldsymbol{x}}_0^{(i)}$. Then, the advantage of the $i$-th sampled trajectory at time $t$ is calculated by normalizing the group-level rewards:

$$\hat{A}_t^{(i)} = \frac{r^{(i)} - \text{mean}(\{r^{(i)}\}_{i=1}^G)}{\text{std}(\{r^{(i)}\}_{i=1}^G)}. \tag{4}$$

Finally, the GRPO algorithm updates the policy model by maximizing the following objective:

$$\mathcal{J}_{\text{DDIM-GRPO}}(\theta) = \mathbb{E}_{\boldsymbol{c}_{\text{LQ}} \sim p(\boldsymbol{c}_{\text{LQ}}), \{\hat{\boldsymbol{x}}^{(i)}\}_{i=1}^G \sim \pi_{\theta_{\text{old}}}(\cdot|\boldsymbol{c}_{\text{LQ}})}$$

$$\frac{1}{G} \sum_{i=1}^G \frac{1}{T} \sum_{t=0}^{T-1} \left[ \min\left( r_t^{(i)}(\theta)\hat{A}_t^{(i)}, \text{clip}\left( r_t^{(i)}(\theta), 1-\epsilon, 1+\epsilon \right) \hat{A}_t^{(i)} \right) - \beta D_{KL}(\pi_\theta \| \pi_{\text{ref}}) \right], \tag{5}$$

where $r_t^{(i)}(\theta) = \frac{\pi_\theta(\boldsymbol{x}_{t-1}^{(i)}|\boldsymbol{x}_t^{(i)}, \boldsymbol{c}_{\text{LQ}})}{\pi_{\theta_{\text{old}}}(\boldsymbol{x}_{t-1}^{(i)}|\boldsymbol{x}_t^{(i)}, \boldsymbol{c}_{\text{LQ}})}$, with $T$ denoting the total timesteps and $\pi_{\text{ref}}$ denotes the initialized pretrained model.

Due to the inherent randomness of the Diffusion model, given the same LQ face $\boldsymbol{c}_{\text{LQ}}$, a set of diverse candidate faces $\{\hat{\boldsymbol{x}}_0^{(1)}, \hat{\boldsymbol{x}}_0^{(2)}, \ldots, \hat{\boldsymbol{x}}_0^{(G)}\}$ can be generated from policy $\pi_\theta$. Ideally, when $\pi_\theta$ learns a posterior distribution capturing different potential solutions, the generated samples cluster around distinct peaks $\{\boldsymbol{\mu}_k\}$ in the solution space. Such diverse candidates enable the optimization process to make meaningful comparisons and drive policy improvements.

Once obtaining the candidate set, we evaluate each generated sample $\hat{\boldsymbol{x}}_0^{(i)}$ using the composite reward function $r(\cdot)$. The GRPO algorithm then transforms the absolute rewards $r^{(i)}$ into within-group relative advantages $\hat{A}_t^{(i)}$ (See Eq. 4), which improves the ability to distinguish between high and low-quality solutions while reducing sensitivity to reward scaling. The core insight of this transformation is to redirecting optimization from absolute quality assessment ("how good is this solution?") to relative comparison ("how does this solution rank within the group?"). Consequently, the policy gradient update direction is proportional to:

$$\sum_{i=1}^G \sum_{t=0}^{T-1} \hat{A}_t^{(i)} \nabla_\theta \log \pi_\theta \left( \boldsymbol{a}_t^{(i)}|\boldsymbol{s}_t^{(i)} \right). \tag{6}$$

Thus, the following conclusion can be drawn: When a sample's reward exceeds the group average, its advantage $\hat{A}_t^{(i)}$ is positive, increasing the policy's selection probability; conversely, below-average rewards yield negative advantages $\hat{A}_t^{(j)}$, reducing selection likelihood for suboptimal solutions. This group sampling approach enables parallel exploration of the policy's learned solution space. To fully exploit GRPO's ability to amplify good solutions while suppressing bad ones, we introduce three key innovations that improve exploration efficiency and enhance face restoration quality.

## 4.2 LIKELIHOOD-REGULARIZED POLICY OPTIMIZATION

**Composite Reward Function.** To effectively guide the optimization of the policy network $\pi_\theta$ and enable it to find a better balance in the complex Perception-Distortion Tradeoff (Blau & Michaeli, 2018), we design a multi-objective composite reward function, $R(\hat{\boldsymbol{x}}_0, \boldsymbol{x}_{\text{GT}})$. This composite function measures the quality of generated face images $\hat{\boldsymbol{x}}_0$ from three complementary perspectives:

- *Human Preference Reward* ($r_{\text{pref}}$): We employ a Face Reward Model (Wu et al., 2025), pre-trained on a human preference dataset, to score the overall realism and naturalness of facial details, ensuring alignment with human aesthetic preferences.
- *Perceptual Quality* ($r_{\text{aq}}$): We incorporate CLIP-IQA (Wang et al., 2023a), a no-reference image quality assessment metric, to objectively measure perceptual quality using knowledge from pre-trained CLIP (Radford et al., 2021) models.
- *Fidelity Reward* ($r_{\text{fid}}$): We formulate a fidelity reward based on feature similarity and wavelet low-frequency constraints to enforce identity consistency, resulting in substantial fidelity improvements (see Appendix C for implementation details).

The final total reward $R(\cdot)$ is defined as the weighted sum of these three components:

$$R(\hat{\boldsymbol{x}}_0, \boldsymbol{x}_{\mathrm{GT}}) = \lambda_1 r_{\mathrm{pref}}(\hat{\boldsymbol{x}}_0) + \lambda_2 r_{\mathrm{aq}}(\hat{\boldsymbol{x}}_0) + \lambda_3 r_{\mathrm{fid}}(\hat{\boldsymbol{x}}_0, \boldsymbol{x}_{\mathrm{GT}}) \tag{7}$$

where $\lambda_{(\cdot)}$ are the weight coefficients for each term.

**Ground-truth Guided Likelihood Regularization.** Optimization based solely on reward maximization is susceptible to reward hacking–the policy may exploit reward function biases to produce unrealistic outputs. We address this with ground-truth (GT) guided likelihood regularization $\mathcal{R}_{\mathrm{likelihood}}$, activated only in the final $S$ timesteps, to maintain realistic and natural face generation.

The regularization leverages GT supervision by using ideal denoising trajectories derived from $\hat{\boldsymbol{x}}^{\mathrm{GT}}$. For each GT image, we pre-compute the ideal latent trajectory $\{\boldsymbol{x}^{\mathrm{GT}}\}_{t=0}^{T}$ through forward noising. During training, when the policy $\pi_\theta$ samples a state $\boldsymbol{x}_t$, we apply regularization by maximizing the log-likelihood of the model producing the ideal subsequent state $\boldsymbol{x}_{t-1}^{GT}$. Thus, the regularization item is defined as:

$$\mathcal{R}_{\mathrm{likelihood}} = -\log \pi_\theta(\boldsymbol{x}_{t-1}^{\mathrm{GT}}|\boldsymbol{x}_t, \boldsymbol{c}_{\mathrm{LQ}}) \propto \frac{\|\boldsymbol{x}_{t-1}^{\mathrm{GT}} - \mu_\theta(\boldsymbol{x}_t, t, \boldsymbol{c}_{\mathrm{LQ}})\|_2^2}{2\sigma_t^2}. \tag{8}$$

Where $\mu_\theta$ and $\sigma_t$ refer to the mean and standard deviation of DDIM's one-step denoising, with the detailed form provided in Sec. 3. As shown in Eq. 8, this loss term encourages the policy network $\pi_\theta$ to predict a high-probability distribution centered around the ideal target $\boldsymbol{x}_{t-1}^{\mathrm{GT}}$ at its explored state $\boldsymbol{x}_t$. This is equivalent to minimizing the variance-weighted Euclidean distance between the predicted mean $\mu_\theta(\boldsymbol{x}_t, t, \boldsymbol{c})$ and the ideal state $\boldsymbol{x}_{t-1}^{\mathrm{GT}}$. The likelihood regularization $\mathcal{R}_{\mathrm{likelihood}}$ maintains alignment between restored image and authentic facial distributions, substituting for the standard KL divergence term in GRPO.

**Noise-Level Advantage Assignment.** Conventional GRPO-based approaches (Liu et al., 2025; Fan et al., 2023) treat all timesteps equally when assigning advantage weights, ignoring the inherently non-uniform importance of different steps in the diffusion process. To address this, inspired by previous work (He et al., 2025), we introduce a noise-level-aware advantage assignment approach that correlates advantage weights with the exploration magnitude achieved at each denoising step. In DDIM sampling, the single-step exploration radius from $\boldsymbol{x}_t$ to $\boldsymbol{x}_{t-1}$ is determined by the standard deviation $\sigma_t$ of the added noise. Therefore, we set the timestep weight $w_t$ proportional to $\sigma_t$:

$$w_t \propto \sigma_t, \quad \text{s.t.} \quad \frac{1}{T} \sum_{t=0}^{T-1} w_t = 1 \tag{9}$$

The specific form of $\sigma_t$ is detailed in Sec. 3. After normalizing the weights $\{w_t\}_{t=1}^{T}$, we apply them to weight the original advantage $\hat{A}_t^{(i)}$, yielding the final advantage $\tilde{A}_t^{(i)}$ for policy update:

$$\tilde{A}_t^{(i)} = w_t \cdot \hat{A}_t^{(i)} \tag{10}$$

Since $\sigma_t$ decreases from high initial values to nearly zero, early denoising steps possess greater exploration capability. By weighting these steps more heavily in advantage computation, we facilitate enhanced exploration that yields more diverse high-quality restoration outcomes. Simultaneously, it reduces interference during the high-frequency detail refinement that occurs in later denoising phases. This allocation strategy effectively performs weighted adjustment of policy gradients, as detailed in Appendix A.

**LRPO Optimization Objective.** By combining these strategies, we formulate an optimization objective that maximizes policy return while applying likelihood regularization to prevent unrealistic restorations:

$$\mathcal{J}_{\mathrm{LRPO}}(\theta) = \mathbb{E}_{\boldsymbol{c}_{\mathrm{LQ}} \sim p(\boldsymbol{c}_{\mathrm{LQ}}), \{\hat{\boldsymbol{x}}^{(i)}\}_{i=1}^{G} \sim \pi_{\theta_{\mathrm{old}}}(\cdot|\boldsymbol{c}_{\mathrm{LQ}})}$$

$$\frac{1}{G} \sum_{i=1}^{G} \frac{1}{T} \sum_{t=0}^{T-1} \left[ \min\left( r_t^{(i)}(\theta)\hat{A}_t^{(i)}, \mathrm{clip}\left( r_t^{(i)}(\theta), 1-\epsilon, 1+\epsilon \right) \hat{A}_t^{(i)} \right) + \alpha \mathcal{R}_{\mathrm{likelihood}}^{(i)} \right], \tag{11}$$

Compared to Eq. 5, we replace the KL divergence component with GT-guided likelihood regularization in our optimization objective.

## 5 EXPERIMENTS

### 5.1 EXPERIMENTAL SETTINGS

Experimental hyperparameters are detailed in Appendix B, and composite reward function configurations are provided in Appendix C.

**Training and Testing Data.** We only use 3000 face images from the FFHQ (Karras et al., 2021) dataset for training. The degradation strategy from HQ to LQ is based on the following degradation function: $\mathbf{I}_{\text{LQ}} = \left\{ \left[ \left( \mathbf{I}_{\text{HQ}} \otimes \boldsymbol{k}_\sigma \right)_{\downarrow_r} + \boldsymbol{n}_\delta \right]_{\text{JPEG}_q} \right\}_{\uparrow_r}$, where the HQ images are first convolved with a Gaussian kernel $\boldsymbol{k}_\sigma$, followed by a downsampling with a factor of $r$, and then corrupted with Gaussian noise $\boldsymbol{n}_\delta$. Subsequently, the images undergo JPEG compression with a quality factor of $q$. Finally, the LQ image is resized back to the original $512 \times 512$. Here, $\sigma$, $r$, $\delta$, and $q$ are randomly sampled from the intervals $[0.1, 12]$, $[1, 12]$, $[0, 15]$, and $[30, 100]$, respectively. Follow previous work Wang et al. (2021); Gu et al. (2022), we employ the synthetic dataset CelebA-Test (Karras et al., 2017) and two real-world datasets (Wang et al., 2021) (*i.e.*, LFW-Test and WebPhoto-Test) to validate our method.

**Evaluation Metrics.** On the Celeba-Test dataset, we utilized six common reference-based metrics: SSIM (Wang et al., 2004), PSNR, LPIPS (Zhang et al., 2018), CLIP Score (Hessel et al., 2021), Deg. (Wang et al., 2021), and LMD (Gu et al., 2022), where Deg. and LMD are identity consistency metrics, along with four non-reference metrics: MUSIQ (Ke et al., 2021), MANIQA (Yang et al., 2022), CLIPIQA (Wang et al., 2023a), and Aesthetic (LAION-AI, 2022).

**Comparison Methods.** We compare with not only the base models but also the latest state-of-the-art methods, including GFPGAN (Chan et al., 2021), CodeFormer (Zhou et al., 2022), VQFR (Gu et al., 2022), DR2+SPAR (Wang et al., 2023b), RestoreFormer (Wang et al., 2022), DifFace (Yue & Loy, 2024), OSEDiff (Wu et al., 2024), DiffBIR (Lin et al., 2024), FlowIE (Zhu et al., 2024) and InterLCM (Li et al., 2025).

Table 1: Performance comparisons on CelebA-Test. The highest score for each metric is highlighted in red, the second-highest in blue. Metrics with ↑ indicate higher is better, ↓ means lower is better. Values in parentheses represent our method's improvements over the base DiffBIR model.

| Methods | SSIM↑ | PSNR↑ | LPIPS↓ | CLIP Score↑ | Deg.↓ | LMD↓ | MUSIQ↑ | MANIQA↑ | Aesthetic↑ | CLIPIQA↑ |
|---|---|---|---|---|---|---|---|---|---|---|
| Input | 0.6994 | 25.33 | 0.4866 | 0.7894 | 47.94 | 3.7560 | 17.00 | 0.3957 | 4.0484 | 0.2957 |
| GFP-GAN | 0.6772 | 24.65 | 0.3646 | 0.8410 | 34.58 | 2.4110 | 73.90 | 0.6522 | 5.6992 | 0.6781 |
| CodeFormer | 0.6925 | 25.85 | 0.3335 | 0.8931 | 31.08 | 1.9963 | 74.23 | 0.6520 | 5.8103 | 0.6493 |
| VQFR | 0.6654 | 23.76 | 0.3557 | 0.8562 | 42.48 | 2.9444 | 73.84 | 0.6544 | 5.7844 | 0.6750 |
| DR2+SPAR | 0.6512 | 22.89 | 0.4146 | 0.7437 | 57.24 | 4.5449 | 70.19 | 0.6374 | 5.6602 | 0.5960 |
| DifFace | 0.6762 | 24.80 | 0.3994 | 0.8380 | 45.81 | 2.9766 | 68.96 | 0.6204 | 5.4708 | 0.5711 |
| OSEDiff | 0.6864 | 23.96 | 0.3478 | 0.7962 | 46.20 | 2.8871 | 73.41 | 0.6560 | 5.7720 | 0.6120 |
| FlowIE | 0.6769 | 24.85 | 0.3442 | 0.8961 | 33.44 | 2.1995 | 74.08 | 0.6720 | 5.6782 | 0.6866 |
| InterLCM | 0.6819 | 24.88 | 0.3349 | 0.8905 | 33.58 | 2.1519 | 75.16 | 0.6781 | 5.7735 | 0.6748 |
| DiffBIR | 0.6775 | 25.44 | 0.3811 | 0.8877 | 35.16 | 2.2661 | 74.46 | 0.6752 | 5.7943 | 0.7200 |
| LRPO (ours) | 0.7021 | 26.15 | 0.3635 | 0.9100 | 31.19 | 1.9533 | 75.24 | 0.6808 | 5.8126 | 0.8061 |
| | (+0.0246) | (+0.71) | (+0.0176) | (+0.0223) | (+3.97) | (+0.3128) | (+0.78) | (+0.0056) | (+0.0183) | (+0.0861) |

### 5.2 MAIN RESULTS

**Results on Synthetic Data.** As shown in Table 1, LRPO achieves improvements on all metrics compared with DiffBIR on the synthetic CelebA-Test dataset. These results indicate that our RL framework simultaneously improves perceptual quality and identity preservation in restored faces. Furthermore, LRPO achieves superior performance compared to state-of-the-art approaches across the majority of evaluation metrics, including SSIM, LMD, and MUSIQ, confirming that it enhances identity consistency while maintaining perceptual quality. Figure 3 demonstrates LRPO's superior performance over methods that fail to restore faces satisfactorily. LRPO delivers more realistic textures than the baseline, better identity alignment than DR2 and OSEDiff, and more natural results without the over-smoothing seen in InterICM and FlowIE.

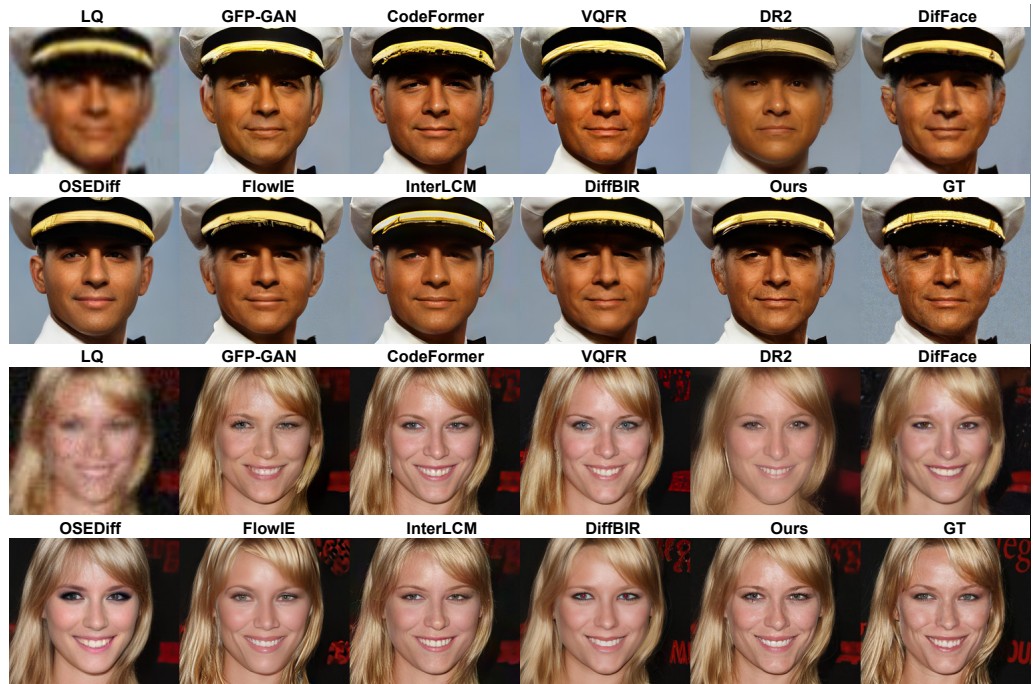

Figure 3: Qualitative results on CelebA-Test datasets. (Zoom in for details)

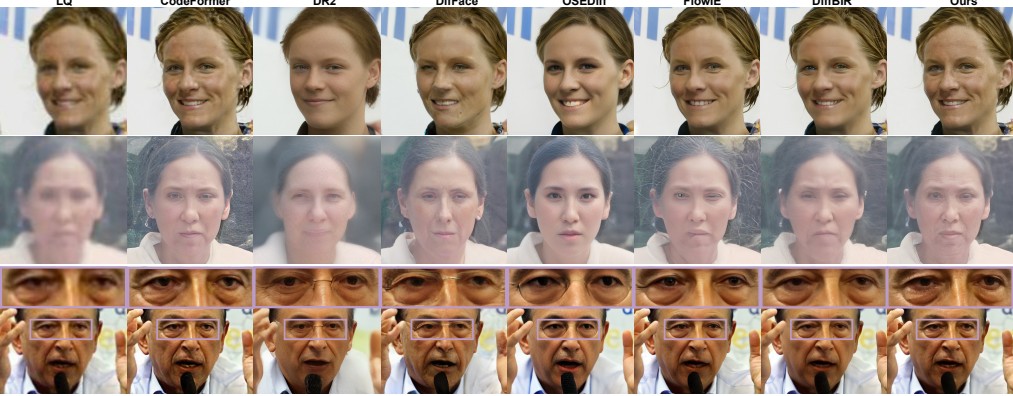

Figure 4: Qualitative results on real-world datasets. (Zoom in for details)

**Results on Real-world Data.** Table 2 shows the quantitative performance evaluation on real-world datasets LFW-Test and WebPhoto-Test. LRPO demonstrates significant performance gains compared to the base DiffBIR and outperforms other state-of-the-art approaches on MUSIQ and CLIP-IQA metrics. Qualitative results are illustrated in Figure 4. Due to severe degradation in real-world inputs, many approaches fail to restore texture details. In contrast, our method recovers more details while introducing fewer artifacts.

**Human Preference Evaluation.** A user study was conducted with 12 participants of varying backgrounds to evaluate 100 face images from the CelebA-Test dataset. Participants evaluated our method against the base model (DiffBIR) on two criteria: *fidelity* (identity preservation) and

Table 2: Performance comparisons on wild datasets. The highest score is highlighted in red, and the second-highest in blue. Metrics with ↑ indicate higher is better.

| Dataset | LFW-Test | | WebPhoto-Test | |
|---|---|---|---|---|
| Methods | MUSIQ↑ | CLIPIQA↑ | MUSIQ↑ | CLIPIQA↑ |
| Input | 26.87 | 0.2834 | 18.63 | 0.4128 |
| GFP-GAN | 73.57 | 0.6983 | 72.09 | 0.6888 |
| CodeFormer | 70.69 | 0.6335 | 71.16 | 0.6573 |
| VQFR | 74.39 | 0.7100 | 70.91 | 0.6767 |
| DR2+SPAR | 72.22 | 0.6427 | 63.65 | 0.5586 |
| DiffFace | 69.85 | 0.6110 | 65.21 | 0.5821 |
| OSEDiff | 73.40 | 0.6327 | 72.60 | 0.6454 |
| FlowIE | 64.29 | 0.5974 | 71.45 | 0.6838 |
| InterLCM | 74.18 | 0.6588 | 73.91 | 0.6658 |
| DiffBIR | 73.71 | 0.7296 | 67.45 | 0.6630 |
| LRPO (ours) | 74.60 (+0.89) | 0.8073 (+0.0777) | 72.71 (+5.26) | 0.7040 (+0.0410) |

*realism* (naturalness with minimal artifacts). As shown in Table 3, our method outperforms the base model in both fidelity and realism according to human preferences.

## 5.3 ABLATION STUDY

We conduct the ablation study on CelebA-Test dataset. As shown in Table 4, we analyze the effects of four key components: Reinforcement Learning (RL), Kullback-Leibler divergence (KL), GT guided likelihood regularization (Reg.), and noise-level advantage assignment (AdA). Variant 1 demonstrates improvements across all metrics after incorporating RL, confirming that RL directly enhances BFR performance. However, adding KL divergence in Variant 2 degrades all metrics without improving visual quality (See Figure 5(a)). We therefore remove KL divergence from the RL objective in Eq. 11, reducing computational cost while maintaining visual quality. Without AdA (Variant 3), the model suffers from detail blurring caused by over-optimization during late denoising stages (Figure 5(b)). Removing Reg. (Variant 4) leads to decreased SSIM scores and poor fidelity, with the model producing unrealistic, fantasy-like textures as shown in Figure 5(c). Training dynamics show that our noise-level advantage assignment facilitates faster convergence to high-reward restoration trajectories, while GT guided likelihood regularization enhances structural fidelity (Appendix D).

Table 3: User study. Participants selected the winner between DiffBIR and LRPO restored images in terms of fidelity and realism.

| Comparison | Fidelity % | Realism % |
|---|---|---|
| DiffBIR vs LRPO | 38.1% vs 61.9% | 27.6% vs 72.4% |

Table 4: Ablation Study of LRPO

| Struct | RL | KL | Reg | AdA | SSIM↑ | LMD↓ | CLIPIQA↑ |
|---|---|---|---|---|---|---|---|
| Base | | | | | 0.6775 | 2.2661 | 0.7200 |
| Variant 1 | ✓ | | | | 0.6849 | 2.0503 | 0.7809 |
| Variant 2 | ✓ | ✓ | | | 0.6750 | 2.1602 | 0.7816 |
| Variant 3 | ✓ | | ✓ | | 0.6867 | 2.0078 | 0.7852 |
| Variant 4 | ✓ | | | ✓ | 0.6806 | 1.9551 | 0.7980 |
| LRPO | ✓ | | ✓ | ✓ | **0.7021** | **1.9533** | **0.8061** |

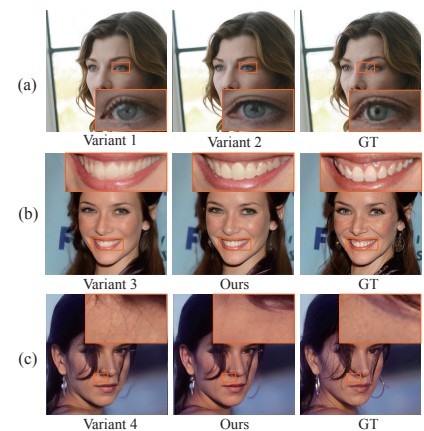

Figure 5: Ablation study visualizations.

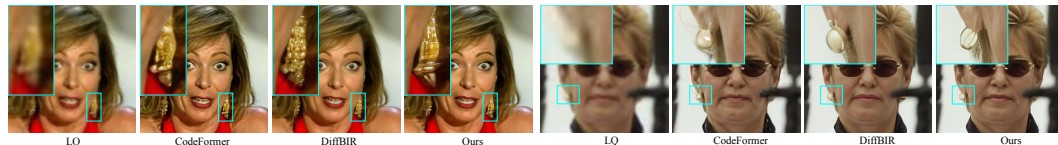

Figure 6: Failure cases. Restoration of highly rare specialized and individualized objects such as jewelry achieves suboptimal results.

## 6 CONCLUSION

In this work, we propose LRPO, the first online reinforcement learning framework applied to BFR tasks. LRPO exploits RL's inherent exploration mechanisms to overcome the limitations of deterministic restoration methods, simultaneously improving perceptual quality and identity preservation. LRPO integrates three critical innovations: a composite reward function for multi-perspective image evaluation, GT guided likelihood regularization for fidelity preservation, and noise-level advantage assignment for efficient optimization. Comprehensive experiments validate LRPO's effectiveness in enhancing both identity consistency and perceptual quality compared to existing approaches.

**Limitation.** While our method surpasses existing approaches, certain failure cases remain. Figure 6 demonstrates that rare specialized and individualized objects (*e.g.*, jewelry) are restored with artifacts due to limited prior knowledge in the base model's training data. These limitations may require stronger foundation models or more comprehensive training datasets.

## ETHICS STATEMENT

All authors adhere to the ICLR Code of Ethics. Our research is confined to the technical challenge of image restoration and does not introduce new ethical risks. By improving identity consistency, our method aims to mitigate known issues in face restoration. All experiments were conducted using publicly available datasets for training and evaluation.

## REPRODUCIBILITY STATEMENT

To ensure reproducibility, our source code and models will be publicly released. Our experiments use public datasets. All implementation details, training hyperparameters, and the composition of our reward function are provided in Appendix.

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

## A  ADVANCE WEIGHT AS A DIRECT GRADIENT COEFFICIENT

For a generative process parameterized by $\theta$, the policy gradient in the DDIM optimization objective can be expressed as follows:

$$\nabla_\theta \mathcal{J}_{\text{LRPO}}(\theta) = \sum_{t=0}^{T-1} \mathbb{E}_{\boldsymbol{x}_T, \boldsymbol{\epsilon}} \left[ \nabla_\theta \log \pi_\theta \left( \boldsymbol{x}_{t-1} | \boldsymbol{x}_t, \boldsymbol{c} \right) \left( w_t \cdot \hat{A}_t \right) \right] \tag{12}$$

The core of the proof is to expand the log-policy gradient term, $\nabla_\theta \log \pi_\theta$. The log-policy is defined by the Gaussian sampling step:

$$\log \pi_\theta(\boldsymbol{x}_{t-1} | \boldsymbol{x}_t, \boldsymbol{c}) = -\frac{\|\boldsymbol{x}_{t-1} - \mu_\theta(\boldsymbol{x}_t, t, \boldsymbol{c})\|_2^2}{2\sigma_t^2} + \mathcal{C}_t \tag{13}$$

Taking the gradient with respect to $\theta$:

$$\nabla_\theta \log \pi_\theta(\boldsymbol{x}_{t-1} | \boldsymbol{x}_t, \boldsymbol{c}) = \nabla_\theta \left( -\frac{\|\boldsymbol{x}_{t-1} - \mu_\theta(\boldsymbol{x}_t, t, \boldsymbol{c})\|_2^2}{2\sigma_t^2} \right) \tag{14}$$

$$= -\frac{1}{2\sigma_t^2} \cdot 2(\boldsymbol{x}_{t-1} - \mu_\theta(\boldsymbol{x}_t, t, \boldsymbol{c})) \cdot (-\nabla_\theta \mu_\theta(\boldsymbol{x}_t, t, \boldsymbol{c})) \tag{15}$$

$$= \frac{\boldsymbol{x}_{t-1} - \mu_\theta(\boldsymbol{x}_t, t, \boldsymbol{c})}{\sigma_t^2} \cdot \nabla_\theta \mu_\theta(\boldsymbol{x}_t, t, \boldsymbol{c}) \tag{16}$$

Since $\boldsymbol{x}_{t-1} = \mu_\theta(\boldsymbol{x}_t, t, \boldsymbol{c}) + \sigma_t \boldsymbol{\epsilon}$ where $\boldsymbol{\epsilon} \sim \mathcal{N}(\mathbf{0}, \mathbf{I})$:

$$\nabla_\theta \log \pi_\theta(\boldsymbol{x}_{t-1} | \boldsymbol{x}_t, \boldsymbol{c}) = \frac{\sigma_t \boldsymbol{\epsilon}}{\sigma_t^2} \cdot \nabla_\theta \mu_\theta(\boldsymbol{x}_t, t, \boldsymbol{c}) = \frac{\boldsymbol{\epsilon}}{\sigma_t} \cdot \nabla_\theta \mu_\theta(\boldsymbol{x}_t, t, \boldsymbol{c}) \tag{17}$$

Expanding $\nabla_\theta \mu_\theta(\boldsymbol{x}_t, t, \boldsymbol{c})$:

$$\nabla_\theta \mu_\theta(\boldsymbol{x}_t, t, \boldsymbol{c}) = \nabla_\theta \left[ \frac{\sqrt{\alpha_{t-1}}}{\sqrt{\alpha_t}} \boldsymbol{x}_t - \frac{\sqrt{\alpha_{t-1}(1-\alpha_t)}}{\sqrt{\alpha_t}} \epsilon_\theta(\boldsymbol{x}_t, t, \boldsymbol{c}) + \sqrt{1-\alpha_{t-1}-\sigma_t^2} \cdot \epsilon_\theta(\boldsymbol{x}_t, t, \boldsymbol{c}) \right] \tag{18}$$

$$= \nabla_\theta \left[ \left( \sqrt{1-\alpha_{t-1}-\sigma_t^2} - \frac{\sqrt{\alpha_{t-1}(1-\alpha_t)}}{\sqrt{\alpha_t}} \right) \epsilon_\theta(\boldsymbol{x}_t, t, \boldsymbol{c}) \right] \tag{19}$$

$$= \left( \sqrt{1-\alpha_{t-1}-\sigma_t^2} - \frac{\sqrt{\alpha_{t-1}(1-\alpha_t)}}{\sqrt{\alpha_t}} \right) \cdot \nabla_\theta \epsilon_\theta(\boldsymbol{x}_t, t, \boldsymbol{c}) \tag{20}$$

Let $C_t$ denote the scalar coefficient in parentheses that varies with timestep $t$. Now, substituting the expansion of $\nabla_\theta \mu_\theta$ back into Eq. 17, we establish the direct relationship:

$$\nabla_\theta \log \pi_\theta(\boldsymbol{x}_{t-1}|\boldsymbol{x}_t, \boldsymbol{c}) = \underbrace{\frac{\boldsymbol{\epsilon}}{\sigma_t} C_t}_{K_t} \cdot \nabla_\theta \epsilon_\theta(\boldsymbol{x}_t, t, \boldsymbol{c}) \tag{21}$$

Here, we have explicitly derived the coefficient $K_t$. Finally, we substitute this complete form into the LRPO gradient objective:

$$\nabla_\theta \mathcal{J}_{\text{LRPO}}(\theta) = \sum_{t=0}^{T-1} \mathbb{E}_{\boldsymbol{x}_T, \boldsymbol{\epsilon}} \left[ (K_t \cdot \nabla_\theta \epsilon_\theta(\boldsymbol{x}_t, t, \boldsymbol{c})) (w_t \cdot \hat{A}_t) \right] \tag{22}$$

$$= \sum_{t=0}^{T-1} \mathbb{E}_{\boldsymbol{x}_T, \boldsymbol{\epsilon}} \left[ (w_t K_t) \cdot \hat{A}_t \cdot \nabla_\theta \epsilon_\theta(\boldsymbol{x}_t, t, \boldsymbol{c}) \right] \tag{23}$$

This formally proves that our advantage weight, $w_t$, becomes part of a new scalar term $(w_t K_t)$ that directly multiplies the network's gradient, $\nabla_\theta \epsilon_\theta$. More importantly, this derivation reveals how our noise-level advantage assignment directly translates into a principled modulation of the learning signal at different stages of the denoising process. The final update to the network's parameters $\theta$ is effectively scaled by our time-dependent weight. In the early stages of denoising, where the model needs to vigorously explore and establish the image's overall structure and identity, our mechanism intelligently increases the optimization intensity. This encourages the policy to discover more diverse and high-quality solutions. Conversely, during the later stages, when the image is mostly formed and the task shifts to refining high-frequency details, our method reduces the optimization strength. This prevents large, disruptive updates from corrupting fine textures and ensures a more stable, fine-grained training process that converges smoothly.

## B   IMPLEMENTATION DETAILS

**Training Setup.** We initialize our policy network with the official pre-trained weights of DiffBIR-v1[1], which was pre-trained on the FFHQ dataset. Our entire framework is built upon PyTorch 2.7.0. The training is conducted on three NVIDIA RTX 4090 GPUs and accelerated using the DeepSpeed library. A key component of our online training pipeline is a dedicated reward server, which is deployed on a separate NVIDIA RTX 4090 GPU to efficiently compute and provide reward signals to the policy network.

**Hyperparameters.** The policy network is optimized using the Adam optimizer with a learning rate of $1 \times 10^{-6}$ and a batch-size of 6. For the denoising process, we employ the DDIM sampler. During training, we set $\eta = 1.0$ to introduce stochasticity that encourages exploration, while for inference, we use $\eta = 0.8$ to achieve more deterministic and stable generation. For our LRPO algorithm, we set the number of candidate samples per group to $G = 9$ and the policy update clipping range to $1 \times 10^{-4}$. The GT-guided likelihood regularization is weighted by a coefficient of $\alpha = 0.001$. Crucially, this regularization is only applied during the final $S = 5$ steps of the denoising process. This strategic application prevents the policy's exploration from being overly constrained during the initial, more impactful stages of the reverse process.

---

[1]The source code and weights from https://github.com/XPixelGroup/DiffBIR.

## C    COMPOSITE REWARD FUNCTION DETAILS

This section provides more details on the reward function, $R(\hat{x}_0, x_{\text{GT}})$. The function is engineered to deliver a holistic assessment of restored images by balancing human aesthetic preference, perceptual quality, and fidelity. The total reward score is a weighted aggregation of four components, formulated as:

$$R(\hat{x}_0, x_{\text{GT}}, c_{\text{text}}) = 0.3 \cdot r_{\text{pref}} + 0.1 \cdot r_{\text{aq}} + 0.3 \cdot r_{\text{lpips}} + 0.3 \cdot r_{\text{dwt}}$$

where $\hat{x}_0$ is the restored image, $x_{\text{GT}}$ is the ground-truth image, and $c_{\text{text}}$ is the textual description corresponding to $x_{\text{GT}}$. We elaborate on each component below.

*Human Preference Reward* ($r_{\text{pref}}$). The human preference evaluation we use is based on the previous work (Wu et al., 2025), called the Face Reward Model. Trained on several human preference datasets, it is able to provide face restoration evaluations with high human consistency. The Face Reward Model's input requires both $\hat{x}_0$ and $c_{\text{text}}$, the latter corresponding to the GT face.

*Perceptual Quality Reward* ($r_{\text{aq}}$). This component, $r_{\text{aq}}$, offers a no-reference evaluation of the image's absolute quality. We employ the CLIPIQA (CLIP-based Image Quality Assessment) metric Wang et al. (2023a) from the `pyiqa` library. It assesses overall perceptual quality and realism without requiring a reference image, making it effective for identifying artifacts.

*Fidelity Reward* ($r_{\text{fid}}$). The fidelity reward ensures the restoration remains faithful to the ground-truth. Our implementation uses a composite metric combining LPIPS to constrain perceptual similarity and a DWT-based measure for structural similarity.

*LPIPS.* The Learned Perceptual Image Patch Similarity (LPIPS) (Zhang et al., 2018) metric computes the distance between two images in a deep feature space, which correlates well with perception. As LPIPS is a distance metric (lower is better), we convert it into a similarity reward (higher is better) via the transformation: $r_{\text{lpips}} = 1.0 - \text{LPIPS}(\hat{x}_0, x_{\text{GT}})$.

*DWT.* To maintain consistency between the restored image and the GT image, we employ a Discrete Wavelet Transform (DWT) as a structural constraint. DWT extracts the low-frequency components of the restored image $\hat{x}_0$ and the GT image $\hat{x}_{\text{GT}}$ for the constraint. We leave the high-frequency components unconstrained to allow for more flexible restoration, reduce interference with high-frequency information, and make the generation more vivid. The detailed formulation is as follows:

$$L_{\text{DWT}} = \|\text{DWT}_{\text{LF}}(\hat{x}_0) - \text{DWT}_{\text{LF}}(\hat{x}_{\text{GT}})\|_1$$

Finally, the $L_{\text{DWT}}$ is converted into a reward score using an exponential decay function, which maps the non-negative loss to a score in the range $(0, 1]$:

$$r_{\text{dwt}} = \exp(-15 \cdot L_{\text{DWT}})$$

The scaling factor is an empirically chosen value.

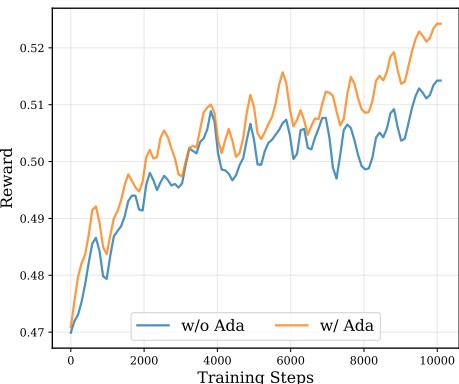

Figure 7: Reward scores during training w/ and w/o noise-level advantage assignment

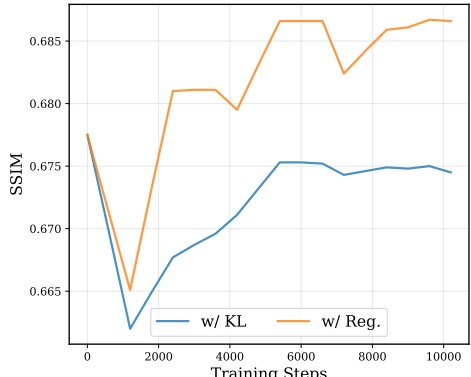

Figure 8: During RL training, the SSIM trend changes are compared between KL and Reg.

## D ABLATION STUDY DETAILS

We provide a more detailed explanation of the four variants presented in Table 4 of the main paper. In Table 4, Variant 2 employs the standard GRPO optimization objective, which includes the KL divergence term. Building upon this, Variant 1 removes the KL term from the optimization objective. While both Variant 3 and Variant 4 follow the GRPO training framework, Variant 3 integrates GT-guided likelihood regularization (Reg.), and Variant 4 integrates noise-level advantage assignment (AdA). Specifically, Variant 3 adds Reg. to the optimization term, with the advantage being assigned uniformly across the time steps. Variant 4 uses the suggested noise-level advantage assignment, but does not incorporate the Reg. term in the optimization objective.

Figure 7 and Figure 8 illustrate the differences in reward and SSIM scores when applying Ada and Reg strategies, respectively. Training dynamics in Figure 7 demonstrate that noise-level advantage assignment consistently outperforms uniform weighting, achieving superior reward scores and faster discovery of optimal restoration solutions. Additionally, Figure 8 validates replacing KL divergence with GT-guided regularization, as evidenced by improved SSIM convergence on CelebA-Test data, indicating better structural alignment with ground truth.

## E THE DETAILS OF HUMAN PREFERENCE EVALUATION

To complement our quantitative metrics, we conducted a human preference evaluation to assess the perceptual quality and fidelity of our proposed LRPO against the DiffBIR. The study involved 12 participants from diverse backgrounds, each evaluating 100 randomly selected face restorations generated by both methods on the CelebA-Test dataset.

For each case, participants were shown the two restored images in a randomized order, along with the corresponding Ground Truth (GT) image for reference. They were then asked to make a forced-choice comparison, selecting one of the two images based on two independent criteria:

- *Realism*: Which image appears more natural and realistic, with richer facial details and fewer visual artifacts?
- *Fidelity*: Which restored face image is more consistent with the identity of the GT face?

Preference rates for realism and fidelity were independently calculated for both methods based on the collected responses. The final results, summarized in Table 3, show that our method was preferred by participants in terms of both realism and fidelity.

## F MORE QUALITATIVE RESULTS

This part shows more quantitative comparisons between our method and others. In Figure 9, we present additional comparison results between our method and others based on the synthetic dataset CelebA-Test. In Figure 10, we present additional comparison results between our method and others based on the real-world datasets.

## G LLM USAGE STATEMENT

In the preparation of this manuscript, we utilized a Large Language Model (LLM) as a writing assistance tool. The role of the LLM was strictly confined to improving the language, including grammar, phrasing, and overall clarity. All scientific contributions, including the core research ideas, experimental methodology, and analysis of results, were developed exclusively by the human authors. The LLM did not contribute to the scientific content of this paper.

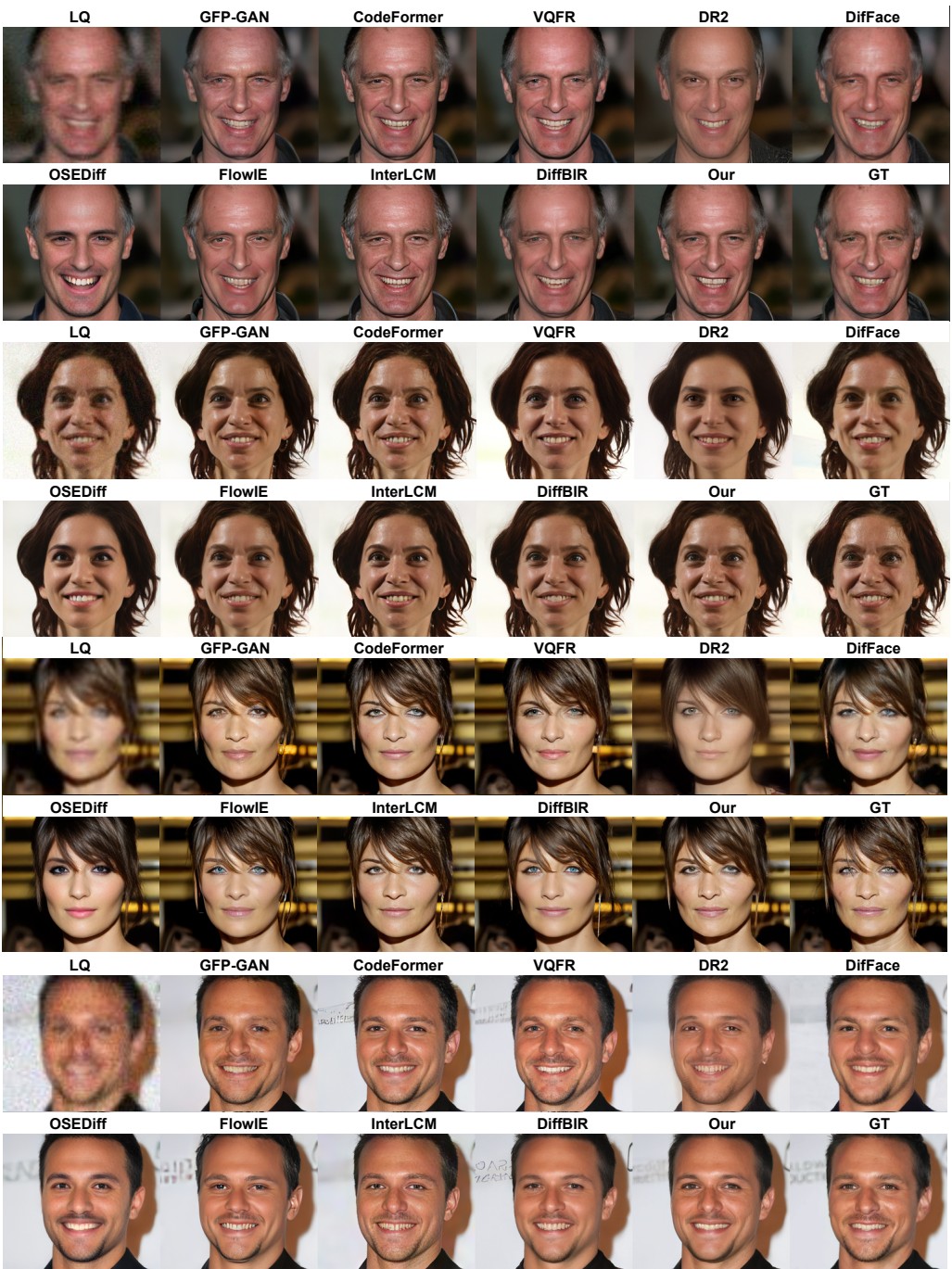

Figure 9: Qualitative results on CelebA-Test datasets. (Zoom in for details)

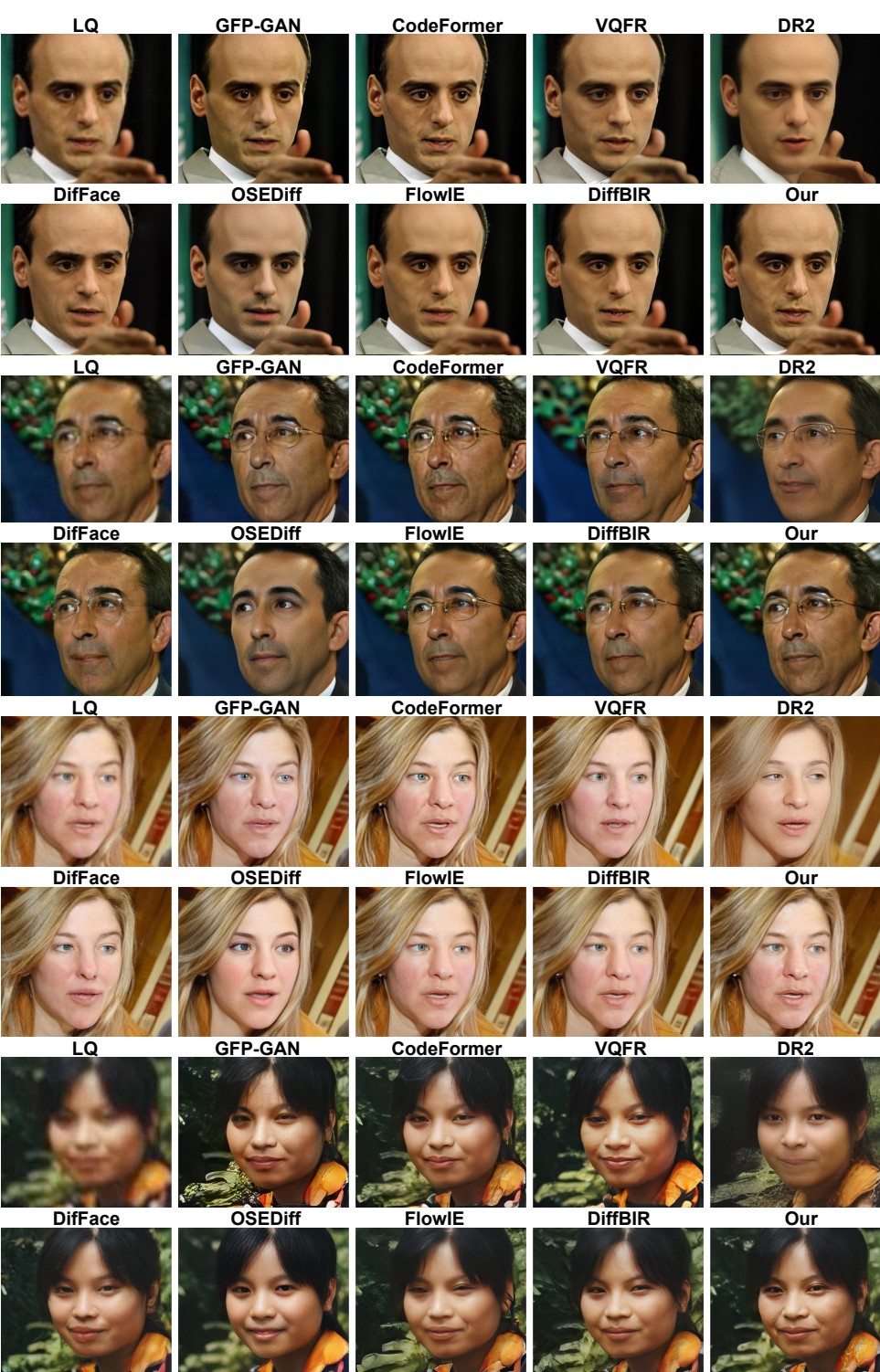

Figure 10: Qualitative results on real-world datasets. (Zoom in for details)

