# OpenReview forum: "LRPO: Enhancing Blind Face Restoration through Online Reinforcement Learning"
_ICLR.cc/2026/Conference — ICLR 2026 Conference Withdrawn Submission_

### Official Review · Reviewer_STrb · 2025-10-24

**Soundness:** 2
**Presentation:** 2
**Contribution:** 2
**Rating:** 4
**Confidence:** 4

**Summary:**

This paper introduces LRPO, a Likelihood-Regularized Policy Optimization framework that applies online reinforcement learning (RL) to the task of blind face restoration (BFR) using diffusion models. The approach incorporates a composite reward function, ground-truth-guided likelihood regularization, and a noise-level advantage assignment to improve image quality, fidelity, and identity preservation in the restored faces. Extensive experiments are conducted on both synthetic and real-world datasets, benchmarking against state-of-the-art BFR methods, and several ablations are performed to assess contribution of each component.

**Strengths:**

1. The work frames BFR as an exploration problem in the solution space using online RL, augmenting diffusion-based face restoration by moving away from deterministic outputs.
2. Development of a multi-faceted reward function balancing human preference, perceptual quality, and fidelity is a mature step toward robust restoration.
3. The proposal to re-weight the RL advantage by noise-level (magnitude of denoising step), informed by the DDIM’s time-dependent stochasticity, is both well-motivated and mathematically justified, as detailed in the appendix and Figure 2 (Page 4).
4. The method outperforms a wide and strong roster of baselines, including modern diffusion- and GAN-based BFR models (see Table 1 & Table 2).

**Weaknesses:**

1. While RL integration for vision (e.g., DDPO, Flow-GRPO, DPOK) has been recently explored, the methodological differences/advantages of LRPO are not always clearly articulated beyond application to BFR. Particularly, the theoretical and empirical gap over DDPO or DPOK is under-analyzed (Related Work, Section 2, lacks direct comparison). The paper mainly shows performance improvement for BFR but does not engage in a critical discussion of what’s distinct about RL for BFR versus other RL-for-diffusion tasks.
2. Training is performed only on FFHQ-derived images (3,000 samples), raising concerns about overfitting or lack of diversity. Given the small dataset, it is unclear if the robustness/generalization extends to “in-the-wild” challenges beyond faces (discussed, e.g., jewelry failures in Figure 6, but not quantitatively explored). Furthermore, reliance on synthetic degradations (for main metrics) may not represent real-world complexity.
3. The reward function uses pretrained models (Face Reward, CLIPIQA, LPIPS+DWT), but the tuning and tradeoff among terms is set empirically (Appendix C), with little justification or sensitivity analysis. The stability of RL training under different reward mixes, and the effect of poor reward calibration on failure modes ("reward hacking") are not sufficiently explored. This limits interpretability and reproducibility when others seek to extend LRPO to other domains.
4. While a broad set of baselines are present (Table 1/2), there is a lack of direct comparison with unsupervised, unpaired, or adaptation-based BFR works (e.g., Kuai et al. [1]), as well as methods leveraging ODEs [2], or advances in multi-prior/fusion (Li et al. [3]). Some directly relevant recent works are not covered in results or discussion, affecting coverage and positioning.
5. In Figure 4 and related results, the observed improvements are less pronounced, and failure cases are mostly qualitative, lacking quantitative error rates.

[1] Kuai T, Honari S, Gilitschenski I, et al. Towards Unsupervised Blind Face Restoration Using Diffusion Prior[C]//2025 IEEE/CVF Winter Conference on Applications of Computer Vision (WACV). IEEE, 2025: 1839-1849.

[2] Van Delft B, Martorella T, Alahi A. CODE: Confident Ordinary Differential Editing[C]//Proceedings of the AAAI Conference on Artificial Intelligence. 2025, 39(20): 20964-20972.

[3] Teng Z, Yu X, Wu C. Blind face restoration via multi-prior collaboration and adaptive feature fusion[J]. Frontiers in Neurorobotics, 2022, 16: 797231.

**Questions:**

1. Please clarify and quantify the sensitivity of the system to the empirical trade-off weights ($\lambda_1, \lambda_2, \lambda_3$) in the reward function. Have you observed mode collapse or reward hacking under different settings?
2. What are the effects of group size $G$ and the number of policy candidates per input? Is performance or stability strongly impacted by adjusting these RL hyperparameters?
3. How robust is LRPO to out-of-distribution degradations or artifacts not seen during training? Have you examined failure rates by attribute (pose, occlusion, expression)?
4. Could the authors elaborate on steps for generalizing the framework beyond faces—for example, blind restoration for other real-world objects or scenes?
5. Why were some of the most recent unsupervised or ODE-based BFR methods not benchmarked as baselines or discussed? Would the authors be able to provide a head-to-head comparison or analysis in revisions?

---

### Official Review · Reviewer_tcGF · 2025-10-27

**Soundness:** 3
**Presentation:** 3
**Contribution:** 3
**Rating:** 6
**Confidence:** 3

**Summary:**

This paper proposes Likelihood-Regularized Policy Optimization, an interesting online reinforcement learning framework for blind face restoration. It seems the first to incorporate RL into this task.
The key contributions are: 1) A composite reward function combining human preference, perceptual quality, and fidelity.  2) A ground-truth-guided likelihood regularization term to prevent reward hacking and maintain realistic facial distributions. 3) A noise-level advantage assignment mechanism to weigh denoising steps based on their significance. Experiments on synthetic and real-world datasets show that LRPO consistently outperforms prior works in both fidelity and perceptual quality metrics. Ablation studies confirm the effectiveness of each proposed component.

**Strengths:**

+ Introducing online RL into diffusion-based BFR is original and potentially impactful.

+ The paper clearly articulates the problem of deterministic diffusion methods being unable to explore the multi-solution nature of BFR.

+ The likelihood regularization and noise-aware advantage weighting are thoughtful additions to stabilize RL training and preserve fidelity.

+ The method achieves consistent quantitative improvements over strong baselines and is validated by a human preference study.

+ Includes both synthetic and real-world data, ablations, and qualitative visualizations that show clear improvement in texture and realism.

**Weaknesses:**

- The framework builds heavily on DiffBIR and inherits its limitations (e.g., bias toward its training data).

- The composite reward function involves multiple pre-trained models (CLIPIQA, Face Reward Model, DWT, LPIPS), which may complicate reproducibility and raise concerns about stability and bias. Does another combination benefit the reward?

- Using only 3,000 FFHQ samples for training seems small for an RL-based system; it’s unclear if overfitting or instability occurs and why not using more FFHQ images for training.

- Online RL with multiple candidate samples per input (G=9) could be computationally heavy, but runtime and convergence details are sparse.

**Questions:**

- How sensitive is LRPO to the choice and weighting of the three reward components (λ1, λ2, λ3)?

- How stable is the RL optimization during training? Are there any divergence or collapse cases observed?

- Could the method generalize to non-face restoration tasks, or is it highly specialized to face priors?

- Why was the KL divergence removed rather than jointly used with likelihood regularization? Does this affect policy exploration behavior?

---

### Official Review · Reviewer_DHpk · 2025-10-27

**Soundness:** 3
**Presentation:** 4
**Contribution:** 3
**Rating:** 4
**Confidence:** 5

**Summary:**

The paper introduces LRPO, a novel online reinforcement learning framework for BFR that explores multiple high-quality candidates in a single diffusion sampling trajectory. By combining a composite reward function, ground-truth-guided likelihood regularization, and noise-level advantage assignment, LRPO achieves state-of-the-art fidelity and perceptual quality in one-shot generation. Extensive experiments on CelebA-Test, LFW-Test and WebPhoto-Test demonstrate consistent gains across SSIM, PSNR, LMD and CLIPIQA metrics.

**Strengths:**

1.	The proposed method is the first to apply online reinforcement learning to the BFR task
2.	The manuscript is well-structured, clearly written, and easy to follow.
3.	The presented qualitative results indicate improvement over several baselines.

**Weaknesses:**

1. Why does the author choose DiffBIR as the baseline? Why not use a dedicated BFR model as the baseline?\
2. How is the size of G defined or determined in line 215? \
3. In Fig. 1 (Right), does “Before” refer to the baseline method DiffBIR used in the paper? \
4. Which parameters are trained in the Policy Network, and what is the rationale behind this choice? \
5. Does LRPO incur additional inference time compared to the baseline DiffBIR? The authors are encouraged to report and compare the inference times of both methods.

**Questions:**

It is unclear why the authors chose DiffBIR as the baseline rather than models specifically designed for BFR. The authors should justify this choice and discuss whether BFR-specific baselines would yield improved results. If this issue is clarified, I will adjust my initial rating

---

### Official Review · Reviewer_7adS · 2025-10-31

**Soundness:** 2
**Presentation:** 2
**Contribution:** 2
**Rating:** 4
**Confidence:** 5

**Summary:**

The authors focus on the problem of face restoration and introduce a framework called LRPO that applies online reinforcement learning (RL) to BFR. Specifically, LRPO leverages a policy network to generate multiple high-quality face candidates from low-quality inputs, evaluates them using a composite reward function, and optimizes the policy network through relative advantages. The framework incorporates ground-truth guided likelihood regularization and noise-level advantage assignment to balance perceptual quality and fidelity. Extensive experiments demonstrate state-of-the-art performance in face restoration quality.

**Strengths:**

1. Using online reinforcement learning in BFR is interesting. LRPO allows for systematic exploration of the solution space, leading to more diverse and higher-quality restorations compared to deterministic methods.

2. LRPO demonstrates improvements over baseline methods and achieves competitive performance on standard evaluation metrics.

**Weaknesses:**

1. The performance of LRPO heavily relies on the pre-trained diffusion model (DiffBIR) used as the base model. The limitations of the base model, such as its training data and architecture, may affect the final restoration quality. For example, the authors mention that specialized objects like jewelry can still be restored with artifacts due to the base model's limited prior knowledge.

2. The authors do not provide a detailed analysis of the training stability and convergence behavior of LRPO. Given the complexity of the RL optimization process and the multiple components involved, ensuring stable and efficient training could be challenging.

3. The human preference evaluation is conducted with a relatively small number of participants (12) and only on the CelebA-Test dataset. This may not fully represent the diverse human perceptions and preferences in different scenarios and populations.

4. In some cases, the visual results of the proposed method are worse than existing methods (see Fig. 3 and Fig. 9 ).

**Questions:**

1. How do the training time and computational cost of LRPO compare with other state-of-the-art face restoration methods, especially those without RL components?

2. Can the authors provide more details on the training stability and convergence behavior of LRPO, such as the number of training iterations required to achieve stable performance and any observed challenges during training?

3. What is the impact of the noise-level advantage assignment mechanism on the performance of LRPO? How does it compare to uniform weighting of advantages across different denoising steps?

4. How does directly fine-tuning DiffBIR using the proposed reward function as an additional loss compare to using reinforcement learning in terms of performance and efficiency for BFR?

---

### Note · Authors · 2025-11-14

I have read and agree with the venue's withdrawal policy on behalf of myself and my co-authors.